# ETNet: Error Transition Network for Arbitrary Style Transfer

**Chunjin Song**[*]
Shenzhen University
songchunjin1990@gmail.com

**Zhijie Wu**[*]
Shenzhen University
wzj.micker@gmail.com

**Yang Zhou**[†]
Shenzhen University
zhouyangvcc@gmail.com

**Minglun Gong**
University of Guelph
minglun@uoguelph.ca

**Hui Huang**[†]
Shenzhen University
hhzhiyan@gmail.com

## Abstract

Numerous valuable efforts have been devoted to achieving arbitrary style transfer since the seminal work of Gatys et al. However, existing state-of-the-art approaches often generate insufficiently stylized results under challenging cases. We believe a fundamental reason is that these approaches try to generate the stylized result in a single shot and hence fail to fully satisfy the constraints on semantic structures in the content images and style patterns in the style images. Inspired by the works on error-correction, instead, we propose a self-correcting model to predict what is wrong with the current stylization and refine it accordingly in an iterative manner. For each refinement, we transit the error features across both the spatial and scale domain and invert the processed features into a residual image, with a network we call Error Transition Network (ETNet). The proposed model improves over the state-of-the-art methods with better semantic structures and more adaptive style pattern details. Various qualitative and quantitative experiments show that the key concept of both progressive strategy and error-correction leads to better results. Code and models are available at `https://github.com/zhijieW94/ETNet`.

## 1   Introduction

Style transfer is a challenging image manipulation task, whose goal is to apply the extracted style patterns to content images. Starting from the works on neural network based generative models [17, 10, 2, 3], the seminal work of Gatys et al. [9] made the first effort to achieve stylization using neural networks with surprising results. The basic assumption they made is that the feature map output by a deep encoder represents the content information, while the correlations between the feature channels of multiple deep layers encode the style info. Following Gatys et al. [9], subsequent works [8, 16, 26, 5, 31] try to address problems in quality, generalization and efficiency through replacing the time-consuming optimization process with feed-forward neural networks.

In this paper, we focus on how to transfer arbitrary styles with one single model. The existing methods achieve this goal by simple statistic matching [13, 21], local patch exchange [6] and their combinations [24, 28]. Even with their current success, these methods still suffer from artifacts, such as the distortions to both semantic structures and detailed style patterns. This is because, when there is large variation between content and style images, it is difficult to use the network to synthesize the stylized outputs in a single step; see e.g., Figure 1(a).

---

[*]Equal contribution, in alphabetic order.
[†]Corresponding authors.

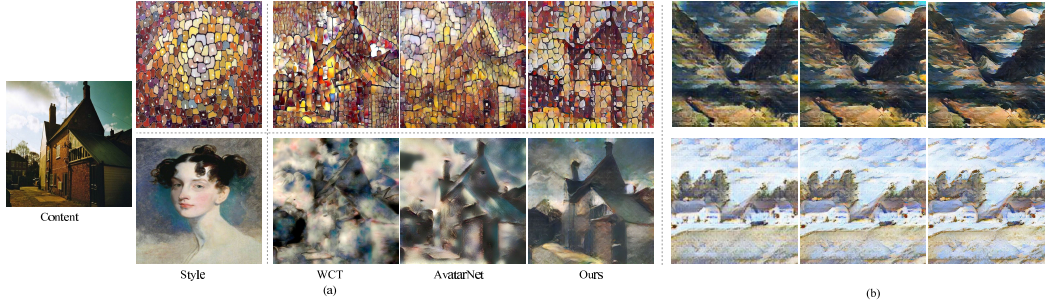

Figure 1: State of the art methods are all able to achieve good stylization with a simple target style (top row in (a)). But for a complex style (bottom row in (a)), both WCT and Avatar-Net distort the spatial structures and fail to preserve texture consistency, our method, however, still performs well. Different from the existing methods, our proposed model achieves style transfer by a coarse-to-fine refinement. One can see that, from left to right in (b), finer details arise more and more along with the refinements. See the close-up views in our supplementary material for a better visualization.

To address these challenges, we propose an iterative error-correction mechanism [11, 22, 4, 18] to break the stylization process into multiple refinements; see examples in Figure 1(b). Given an insufficiently stylized image, we compute what is wrong with the current estimate and transit the error information to the whole image. The simplicity and motivation lie in the following aspect: the detected errors evaluate the residuals to the ground truth, which thus can guide the refinement effectively. Both the stylized images and error features encode information from the same content-style image pairs. Though they do correlate with each other, we cannot simply add the residuals to images or to deep features, especially both on content and style features simultaneously. Thus, a novel error transition module is introduced to predict a residual image that is compatible to the intermediate stylized image. We first compute the correlation between the error features and the deep features extracted from the stylized image. Then errors are diffused to the whole image according to their similarities [27, 14].

Following [9], we employ a VGG-based encoder to extract deep features and measure the distance of current stylization to the ground truth defined by a content-style image pair. Meanwhile, a decoder is designed for error propagation across both the spatial and scale domain and finally invert the processed features back to RGB space. Considering the multi-scale property of style features, our error propagation is also processed in a coarse-to-fine manner. For the coarsest level, we utilize the non-local block to capture long-range dependency, thus the error can be transited to the whole image. Then the coarse error is propagated to finer scale to keep consistency across scales. As a cascaded refinement framework, our successive error-correction procedures can be naturally embedded into a Laplacian pyramid [1], which facilitates both the efficiency and effectiveness for the training.

In experiments, our model consistently outperforms the existing state-of-the-art models in qualitative and quantitative evaluations by a large margin. In summary, our contributions are:

- We first introduce the concept of error-correction mechanism to style transfer by evaluating errors in stylization results and correcting them iteratively.
- By explicitly computing the features for perceptual loss in a feed-forward network, each refinement is formulated as an error diffusion process.
- The overall pyramid-based style transfer framework can perform arbitrary style transfer and synthesize highly detailed results with favored styles.

## 2    Related Work

Under the context of deep learning, many works and valuable efforts [9, 16, 20, 26, 29, 33, 31, 32] have approached the problems of style transfer. In this section, we only focus on the related works on arbitrary style transfer and refer the readers to [15] for a comprehensive survey.

The seminal work of Gatys et al. [9] first proposes to transfer arbitrary styles via a back-propagation optimization. Our approach shares the same iterative error-correction spirit with theirs. The key

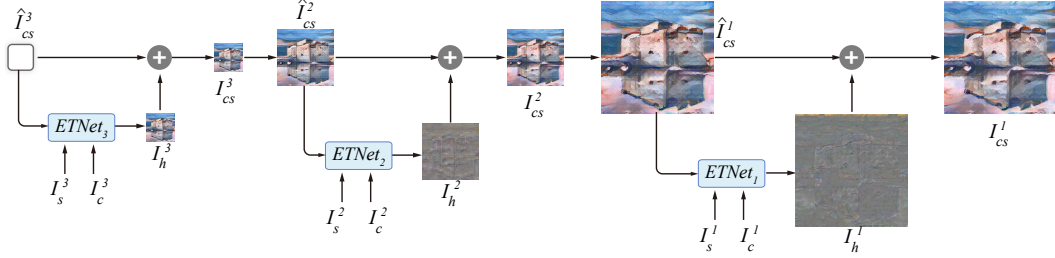

Figure 2: Framework of our proposed stylization procedure. We start with a zero vector to represent the initial stylized image, i.e. $\hat{I}_{cs}^3 = 0$. Together with downsampled input content-style image pair ($I_c^3$ and $I_s^3$), it is fed into the residual image generator $ETNet_3$ to generate a residual image $I_h^3$. The sum of $I_h^3$ and $\hat{I}_{cs}^3$ gives us the updated stylized image $I_{cs}^3$, which is then upsampled into $\hat{I}_{cs}^2$. This process is repeated across two subsequent levels to yield a final stylized image $I_{cs}^1$ with full resolution.

difference is that our method achieves error-correction in a feed-forward manner, which allows us to explicitly perform joint analysis between errors and the synthesized image. With the computed (especially the long-range) dependencies guiding the error diffusion, a better residual image can hence be generated.

Since then, several recent papers have proposed novel models to reduce the computation cost with a feed-forward network. The methods based on, local patch exchange [6], global statistic matching [13, 21, 19] and their combinations [24, 28], all develop a generative decoder to reconstruct the stylized images from the latent representation. Meanwhile, Shen et al. [23] employ the meta networks to improve the flexibility for different styles. However, for these methods, they all stylize images in one step, which limits their capability in transferring arbitrary styles under challenging cases. Hence when the input content and style images differ a lot, they are hard to preserve spatial structures and receive style patterns at the same time, leading to poor stylization results.

Different from the above methods, our approach shares the same spirit with networks coupled with error correction mechanism [11, 22, 4, 18]. Rather than directly learning a mapping from input to target in one step, these networks [30] resolve the prediction process into multiple steps to make the model have an error-correction procedure. The error-correction procedures have many successful applications, including super-resolution [11], video structure modeling [22], human pose estimation [4] and image segmentation [18] and so on. To our knowledge, we make the first efforts to introduce error-correction mechanism to arbitrary style transfer. Coupled with Laplacian pyramid, the proposed schema enables the networks to gradually bring the results closer to the desired stylization and synthesize outputs with better structures and fine-grained details.

## 3 Developed Framework

We formulate style transfer as a successive refinements based on error-correction procedures. As shown in Figure 2, the overall framework is implemented using a Laplacian pyramid [1, 12, 7], which has 3 levels. Each refinement is performed by an error transition process with an affinity matrix between pixels [25, 14] followed by an error propagation procedure in a coarse-to-fine manner to compute a residual image. Each element of the affinity matrix is computed on the similarity between the error feature of one pixel and the feature of stylized result at another pixel, which can be used to measure the possibility of the error transition between them. See Section 3.1 for more details.

### 3.1 Error Transition Networks for Residual Images

We develop Error Transition Network to generate a residual image for each level of a pyramid. As illustrated in Figure 3, it contains two VGG-based encoders and one decoder. Taken a content-style image pair ($I_c$ and $I_s$) and an intermediate stylized image ($\hat{I}_{cs}$) as inputs, one encoder ($\Theta_{in}(\cdot)$) is used to extract deep features $\{f_{in}^i\}$ from $\hat{I}_{cs}$, i.e. $f_{in}^i = \Theta_{in}^i(\hat{I}_{cs})$ and the other encoder ($\Theta_{err}(\cdot)$) is for the error computation. The superscript $i$ represents the feature extracted at the *relu_i_1* layer. The deepest features of both encoders we used are the output of *relu_4_1* layer, which means $i \in \{1, 2, \cdots, L\}$

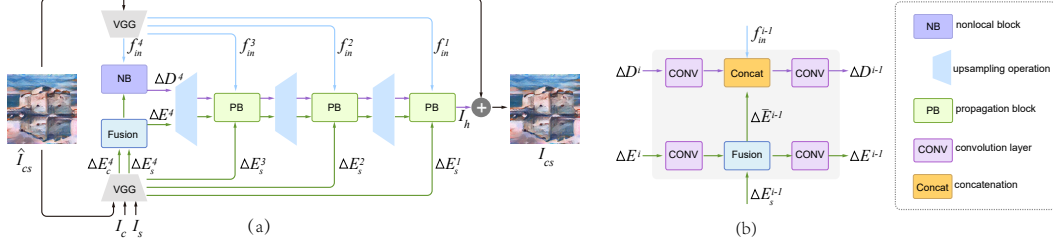

Figure 3: Error Transition network (a) and the detailed architecture of error propagation block (b). For ETNet, the input images include a content-style image pair $(I_c, I_s)$ and an intermediate stylization $(\hat{I}_{cs})$. The two encoders extract deep features $\{f_{in}^i\}$, and error features $\Delta E_c^4$ and $\{\Delta E_s^i\}$ respectively. After the fusion of $\Delta E_c^4$ and $\Delta E_s^4$, we input the fused error feature $\Delta E^4$, together with $f_{in}^4$ into a non-local block to compute a global residual feature $\Delta D^4$. Then both $\Delta D^4$ and $\Delta E^4$ are fed to a series of error propagation blocks to further receive lower-level information until we get the residual image $I_h$. Finally, we add $I_h$ to $\hat{I}_{cs}$ and output the refined image $I_{cs}$.

and $L = 4$. Then a decoder is used to diffuse errors and invert processed features into the final residual image. Following [9], the error of the stylized image $\hat{I}_{cs}$ contains two parts, content error $\Delta E_c^i$ and style error $\Delta E_s^i$, which are defined as

$$\Delta E_c^i(\hat{I}_{cs}, I_c) = \Theta_{err}^i(I_c) - \Theta_{err}^i(\hat{I}_{cs}), \qquad \Delta E_s^i(\hat{I}_{cs}, I_s) = G^i(I_s) - G^i(\hat{I}_{cs}), \qquad (1)$$

where $G^i(\cdot)$ represents a Gram matrix for features extracted at layer $i$ in the encoder network.

To leverage the correlation between the error features and the deep features of stylized image so as to determine a compatible residual feature $\Delta D$, we apply the trick of non-local blocks [27] here. To do that, we first fuse the content and style error features as a full error feature:

$$\Delta E^L = \Psi^L(\Delta E_c^L, \Delta E_s^L) = \Delta E_c^L \cdot W \cdot \Delta E_s^L, \qquad (2)$$

where $\Psi(\cdot)$ and $W$ indicates the fusion operation and a learnable matrix respectively following [33, 31]. Let the $flat(\cdot)$ and $softmax(\cdot)$ denote the flatten operation along the channel dimension and a softmax operation respectively. Then the residual feature at $L$-th scale is determined as:

$$\Delta D^L = flat(\Delta E^L \otimes \psi_h) \cdot softmax(flat(\Delta E^L \otimes \psi_u) \cdot flat(f_{in}^L \otimes \psi_g)^T), \qquad (3)$$

where the $\otimes$ represents one convolution operation and $\{\psi_h, \psi_u, \psi_g\}$ are implemented as learnable $1 \times 1$ convolutions [27]. The affinity matrix outputted by the softmax operation determines the error transition within the whole image. And since long-range dependencies are well captured by the affinity matrix, we are able to better respect the semantic structures, producing superior style pattern consistency. Note that similar to the previous work [27], we only apply the non-local blocks at the top scale (i.e., $L = 4$) to better consider the locality of lower-level features and reduce the computation burden.

Next $\Delta E^L$ and $\Delta D^L$ are fed into cascaded residual propagation blocks to receive lower-level information step by step. As shown in Figure 3(b), a residual propagation block at the $i$-th layer takes four features as input and has two branches which are used to refine $\Delta D^i$ and $\Delta E^i$ respectively. Conditioned on $\Delta E^i$ and $\Delta D^i$ of last scale, to make the residual feature $\Delta D^{i-1}$ more compatible to the stylized image $\hat{I}_{cs}$ and let the fine scale consistent with coarser scales, we take all of them into account to compute residual feature. Thus we achieve $\Delta E^{i-1}$ and $\Delta D^{i-1}$ as:

$$\Delta E^{i-1} = \Delta \bar{E}^{i-1} \otimes \phi_u, \qquad \Delta \bar{E}^{i-1} = \Psi^{i-1}(\Delta E^i \otimes \phi_t, \quad \Delta E_s^{i-1}), \qquad (4)$$

$$\Delta D^{i-1} = [\Delta D^i \otimes \phi_v, \quad f_{in}^{i-1}, \quad \Delta \bar{E}^{i-1}] \otimes \phi_w. \qquad (5)$$

Here $[\cdot, \cdot, \cdot]$ denotes a concatenation operation to jointly consider all inputs for simplicity and $\{\phi_s, \phi_u, \phi_v, \phi_w\}$ represents learnable convolutions for feature adjustment. We further denote $\Delta \hat{E}^i \in R^{N \times C^{i-1}}$ as the processed feature, $\Delta \hat{E}^i = \Delta E^i \otimes \phi_t$. N and $C^{i-1}$ indicate the number of pixels and the channel number of $\Delta \hat{E}^i$ respectively. The learnable matrix $\Psi^{i-1}$ aims to associate $\Delta \hat{E}^i$ and $\Delta E_s^{i-1} \in C^{i-1} \times C^{i-1}$ for additional feature fusion. Both $\Delta E^{i-1}$ and $\Delta D^{i-1}$ will be successively upsampled until $i = 1$. Finally $\Delta D^1$ will be directly outputted as a residual image $I_h$ which will be added to $\hat{I}_{cs}$ to compute $I_{cs}$. Then $I_{cs}$ will be inputted to the finer level of a Laplacian pyramid for further refinement or outputted as the final stylization result.

## 3.2 Style transfer via a Laplacian Pyramid

Figure 2. illustrates a progressive procedure for a pyramid with $K = 3$ to stylize a $768 \times 768$ image. Similar to [7], except the final level, the input images ($I_c^k$ and $I_s^k$) at $k$-th level of the pyramid are downsampled versions of original full resolution images. And correspondingly $\hat{I}_{cs}^k$ denotes the intermediate stylized image. With error transition networks, the stylization is performed in a coarse-to-fine manner. At the begining, we set the initial stylized image $\hat{I}_{cs}^3$ to be an all-zero vector with the same size as $I_c^3$. Setting $k = 3$, together with $I_c^k$ and $I_s^k$, we compute a residual image $I_h^k$ as:

$$I_{cs}^k = \hat{I}_{cs}^k + I_h^k = \hat{I}_{cs}^k + ETNet_k(\hat{I}_{cs}^k, I_c^k, I_s^k), \tag{6}$$

where $ETNet_k$ denotes an error transition network at $k$-th level. Then we upsample $I_{cs}^k$ to expand it to be twice the size and we denote the upsampled version of $I_{cs}^k$ as $\hat{I}_{cs}^{k-1}$. We repeat this process until we go back to the full resolution image. Note that we train each error transition network $ETNet_k(\cdot)$ separately due to the limitation in GPU memory. The independent training of each level also offers benefits on preventing from overfitting [7].

The loss functions are defined based on the image structures present in each level of a pyramid as well. Thus the content and style loss function for $I_{cs}^k$ are respectively defined as:

$$L_{pc}^k = \|\Phi_{vgg}(I_c^k) - \Phi_{vgg}(I_{cs}^k)\|_2 + \sum_{j=k+1}^{K} \|\Phi_{vgg}(I_c^j) - \Phi_{vgg}(\tilde{I}_{cs}^j)\|_2, \tag{7}$$

$$L_{ps}^k = \sum_{i=1}^{L} \|G^i(I_s^k) - G^i(I_{cs}^k)\|_2 + \sum_{j=k+1}^{K} \|G^L(I_s^j) - G^L(\tilde{I}_{cs}^j)\|_2, \tag{8}$$

where the $\Phi_{vgg}$ denotes a VGG-based encoder and as mentioned, we set $L = 4$. And $\tilde{I}_{cs}^j$ is computed as the $(j - k)$ repeated applications of downsampling operation $d(\cdot)$ on $I_{cs}^k$, e.g., $k = 1, \tilde{I}_{cs}^3 = d(d(I_{cs}^1))$ to capture large patterns that can not be evaluated with $\Phi_{vgg}$ directly.

Moreover, total variation loss $L_{TV}(\cdot)$ is also added to achieve the piece-wise smoothness. Thus the total loss at $k$-th level of a pyramid is computed as:

$$L_{total}^k = \lambda_{pc}^k L_{pc}^k + \lambda_{ps}^k L_{ps}^k + \lambda_{tv}^k L_{TV}, \tag{9}$$

where the $\lambda_{pc}^k$ and $\lambda_{tv}^k$ are always set to 1 and $10^{-6}$ while for $k = 1, 2, 3$, $\lambda_{ps}^k$ is assigned to $1, 5, 8$ respectively to preserve semantic structures and gradually add style details to outputs. We tried to use 4 or more levels in the pyramid, but found only subtle improvement achieved on the visual quality.

## 4 Experimental Results

We first evaluate the key ingredients of our method. Then qualitative and quantitative comparisons to several state of the art methods are presented. Finally, we show some applications using our approach, demonstrating the flexibility of our framework. Implementation details and more results can be found in our supplementary document, and code will be made publicly available online.

**Ablation Study**   Our method has three key ingredients: iterative refinements, error measurement and the joint analysis between the error features and features of the intermediate stylization. Table 1 lists the quantitative metrics of the ablation study on above ingredients. For our full model, we can see that though content loss increases a little bit, the style loss shrink significantly by using more refinements, proving the efficacy of our iterative strategy on stylization.

For evaluating the role of error computation, we replace the error features with the plain deep features from the content and style images, i.e. $\Delta E_c^i = \Theta_{err}^i(I_c)$ and $\Delta E_s^i = G^i(I_s)$, before the fusion and information propagation. From the perceptual loss shown in Table 1, the model without error information can still somehow improve the stylization a little bit, since the plain deep features also contain error features, but comparing to our full model of feeding error explicitly, it brings more difficulty for the network to exact the correct residual info. Figure 4 shows a visual comparison. We can see that without error features, noise and artifacts are introduced, like the unseen stripes in the

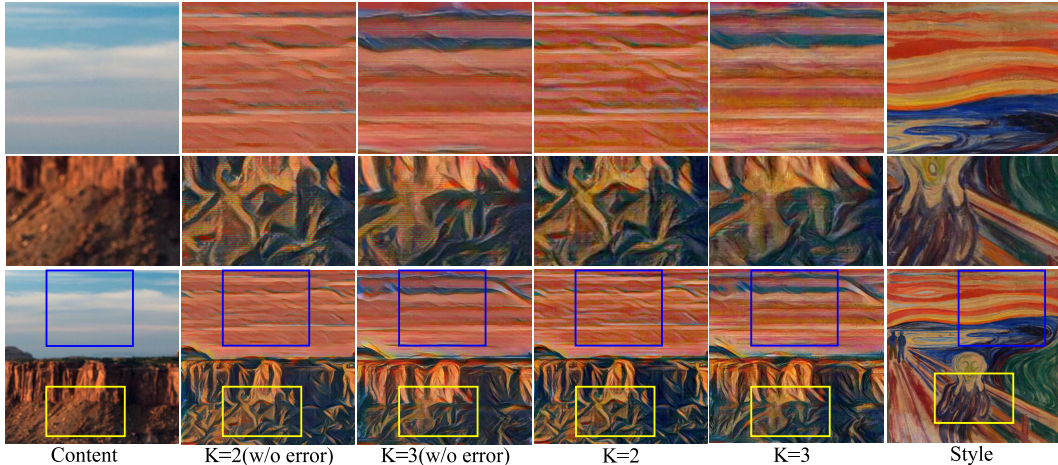

| Content | K=2(w/o error) | K=3(w/o error) | K=2 | K=3 | Style |

Figure 4: Ablation study on explicit error computation. Our full model is successful in reducing artifacts and synthesize texture patterns more faithful to the style image. Better zoom for more details.

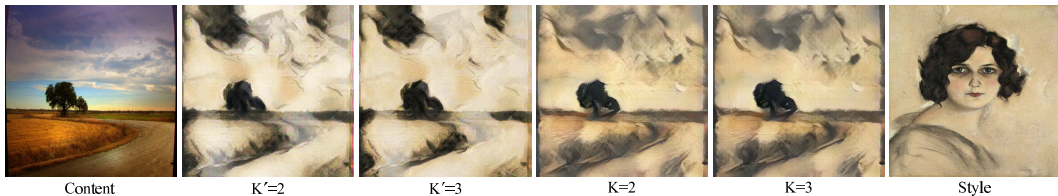

| Content | K′=2 | K′=3 | K=2 | K=3 | Style |

Figure 5: Ablation study on the effect of current stylized results in computing residual images.

Table 1: Ablation study on multiple refinements, the effect of error computation and the joint analysis of intermediate stylized results in computing the residual images. All the results are averaged over 100 synthesized images with perceptual metrics. Note that both $K$ and $K^{'}$ represent the number of refinements, where $K$ denotes a full model and $K^{'}$ represents a simple model that removes the upper encoder, which means it does not consider the intermediate stylization in computing residual images.

| Loss | $K = 1$ | $K = 2$ | $K = 3$ | $K = 1$ w/o err | $K = 2$ w/o err | $K = 3$ w/o err | $K^{'} = 1$ | $K^{'} = 2$ | $K^{'} = 3$ |
|------|---------|---------|---------|------------------|------------------|------------------|-------------|-------------|-------------|
| $L_c$ | 10.9117 | 16.8395 | 18.3771 | 10.9332 | 19.6845 | 19.4235 | 10.9424 | 19.5554 | 22.1014 |
| $L_s$ | 28.3117 | 12.1404 | 7.1984 | 28.3513 | 18.3374 | 15.2981 | 28.3283 | 23.6618 | 21.3698 |

mountain (2nd row, please zoom in to see more details). Our full model yields a much more favorable result. The synthesized patterns faithfully mimic the provided style image.

Then we evaluate the importance of features from the intermediate stylized result in computing the residual image $I_h$. We train a model that removes the upper encoder and directly employ the computed error features to create $I_h$. Specifically, we disable the non-local blocks at the bottleneck layer of our model and for the error propagation block at $i$-th scale, it only takes $\Delta D^i$ and $\Delta E_s^{i-1}$ as inputs. From Table 1, we can see that when disabling the fusion with the features from the intermediate stylization, the final stylizations after iterative refinements are worse than our full model by a large margin. Figure 5 shows an example, where the incomplete model introduces unseen white bumps and blurs the contents more comparing to our full model, demonstrating the effect of the fusion with the intermediate stylization in the error transition.

**Qualitative Comparison** Figure 6 presents the qualitative comparison results to state of the art methods. For baseline methods, codes released by the authors are used with default configurations for a fair comparison. Gatys et a. [9] achieves arbitrary style transfer via time-consuming optimization process but often gets stuck into local minimum with distorted images and fails to capture the salient style patterns. AdaIN [13] frequently produces instylized issues due to its limitation in style

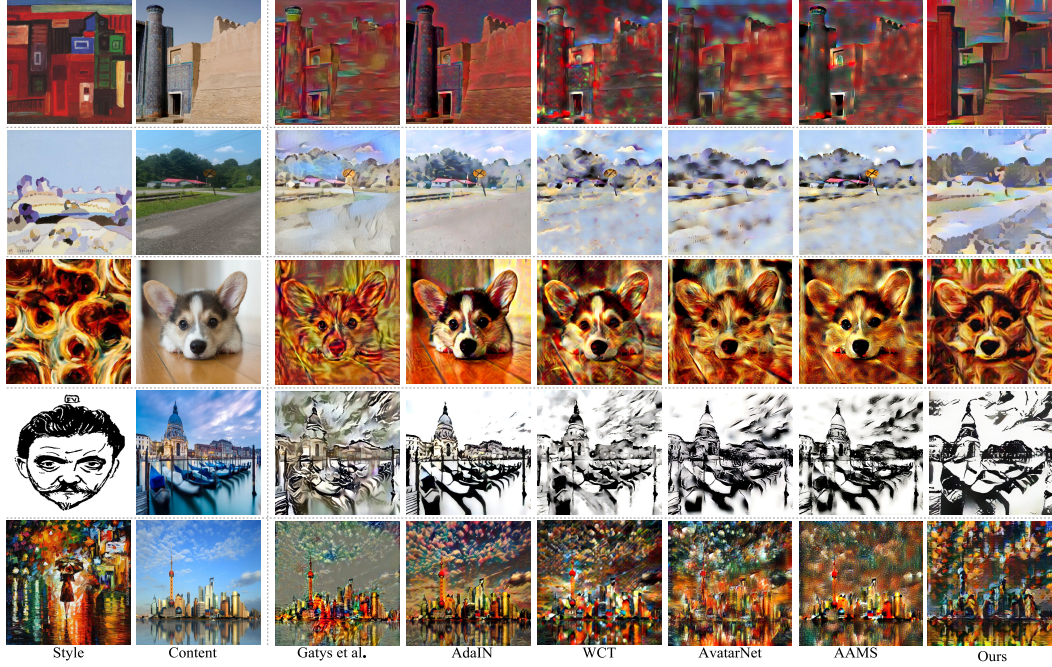

| Style | Content | Gatys et al. | AdaIN | WCT | AvatarNet | AAMS | Ours |

Figure 6: Comparison with results from different methods.

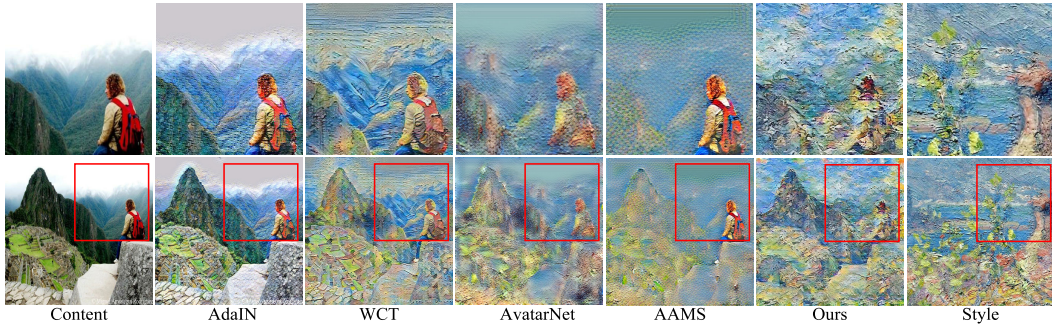

| Content | AdaIN | WCT | AvatarNet | AAMS | Ours | Style |

Figure 7: Detail cut-outs. The top row shows close-ups for highlighted areas for a better visualization. Only our result successfully captures the paint brush patterns in the style image.

representation while WCT [21] improves the generalization ability to unseen styles, but introduces unappealing distorted patterns and warps content structures. Avatar-Net [24] addresses the style distortion issue by introducing a feature decorator module. But it blurs the spatial structures and fails to capture the style patterns with long-range dependency. Meanwhile, AAMS [28] generates results with worse texture consistency and introduces unseen repetitive style patterns. In contrast, better transfer results are achieved with our approach. The iterative refinement coupled with error transition shows a rather stable performance in transferring arbitrary styles. Moreover, the leverage of Laplacian pyramid further helps the preservation of stylization consistency across scales. The output style patterns are more faithful to the target style image, without distortion and exhibiting superior visual detail. Meanwhile, our model better respects the semantic structures in the content images, making the style pattern be adapted to these structures.

In Figure 7, we show close-up views of transferred results to indicate the superiority in generating style details. For the compared methods, they either fail to stylize local regions or capture the salient style patterns. As expected, our approach performs a better style transfer with clearer structures and good-quality details. Please see the brush strokes and the color distribution present in results.

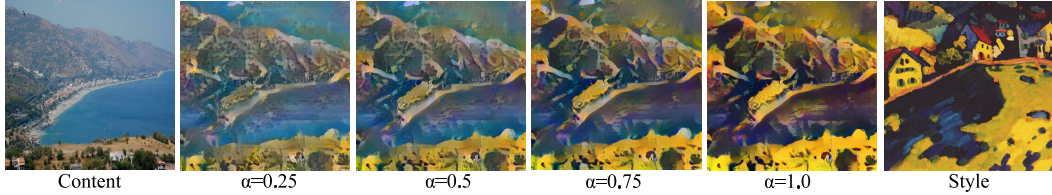

Content  α=0.25  α=0.5  α=0.75  α=1.0  Style

Figure 8: At deployment stage, we can adjust the degree of stylization with paramter $\alpha$.

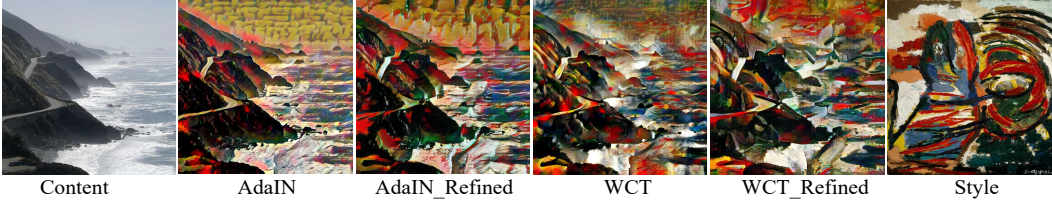

Content  AdaIN  AdaIN_Refined  WCT  WCT_Refined  Style

Figure 9: A refinement for the outputs of AdaIN and WCT.

Table 2: Quantitative comparison over different models on perceptual (content & style) loss, preference score of user study and stylization speed. Note that all the results are averaged over 100 test images except the preference score.

| Loss | AdaIN [13] | WCT [21] | Avatar-Net [24] | AAMS [28] | Ours |
|---|---|---|---|---|---|
| Content($L_c$) | 11.4325 | 19.6778 | 15.5150 | 14.2689 | 18.3771 |
| Style($L_s$) | 71.5744 | 26.1967 | 42.8833 | 40.2651 | 7.1984 |
| Preference/% | 16.1 | 26.4 | 13.0 | 11.2 | 33.3 |
| Time/sec | 0.1484 | 0.7590 | 1.1422 | 1.3102 | 0.5680 |

**Quantitative Comparison**  To quantitatively evaluate each method, we conduct a comparison regarding perceptual loss and report the results in the first two rows of Table 2. It shows that, the proposed method achieves a significant lower style loss than the baseline models, whereas the content loss is also lower than WCT but higher than the other three approaches. This indicates that our model is better at capturing the style patterns presented in style images with good-quality content structures.

It is highly subjective to assess stylization results. Hence, a user study comparison is further designed for the five approaches. We randomly pick 30 content images and 30 style images from the test set and generate 900 stylization results for each method. We randomly select 20 stylized images for each participant who is asked to vote for the method achieving best results. For each round of evaluation, the generated images are displayed in a random order. Finally we collect 600 votes from 30 subjects and detail the preference percentage of each method in the 3rd row of Table 2.

In the 4th row of Table 2, we also compare with the same set of baseline methods [13, 21, 24] in terms of running time. AdaIN [13] is the most efficient model as it uses a simple feed-forward scheme. Due to the requirements for SVD decomposition and patch swapping operations, WCT and Avatar-Net are much slower. Even though the feedback mechanism makes our method slower than the fastest AdaIN algorithm, it is noticeably faster than the other three approaches.

**Runtime applications**  Two applications are employed to reveal the flexibility of designed model at run-time. The same trained model is used for all these tasks without any modification.

Our model is able to control the degree of stylization in running time. For each level $k$ in a pyramid, this can be realized by the interpolation between two kinds of style error features: one is computed between the intermediate output $\hat{I}_{cs}^k$ and style image $I_s^k$ denoting as $\Delta E_{c \to s}$, the other is attained for $\hat{I}_{cs}^k$ and content image $I_c^k$ as $\Delta E_{c \to c}$. Thus the trade-off can be achieved as $\Delta E_{mix} = \alpha \Delta E_{c \to s} + (1 - \alpha) \Delta E_{c \to c}$, which will be fed into the decoder for mixed effect by fusion. Figure 8 shows a smooth transition when $\alpha$ is changed from 0.25 to 1.

With error-correction mechanism, the proposed model is enabled to further refine the stylized results from other existing methods. It can be seen in Figure 9 that, both AdaIN and WCT fail to preserve the global color distribution and introduce unseen patterns. Feeding the output result of these two

methods into our model, ETNet is successful in improving the stylization level by making the color distribution more adaptive to style image and generating more noticeable brushstroke patterns.

## 5    Conclusions

We present ETNet for arbitrary style transfer by introducing the concept of error-correction mechanism. Our model decomposes the stylization task into a sequence of refinement operations. During each refinement, error features are computed and then transitted across both the spatial and scale domain to compute a residual image. Meanwhile long-range dependencies are captured to better resepect the semantic relationships and facilitate the texture consistency. Experiments show that our method significantly improves the stylization performance over existing methods.

## Acknowledgement

We thank the anonymous reviewers for their constructive comments. This work was supported in parts by NSFC (61861130365, 61761146002, 61602461), GD Higher Education Innovation Key Program (2018KZDXM058), GD Science and Technology Program (2015A030312015), Shenzhen Innovation Program (KQJSCX20170727101233642), LHTD (20170003), and Guangdong Laboratory of Artificial Intelligence and Digital Economy (SZ).

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
