[Supplementary Material · ETNet_supp.pdf]

# Supplementary Material of ET-NET: Error Transition Network for Arbitrary Style Transfer

**Chunjin Song**[*]
Shenzhen University
songchunjin1990@gmail.com

**Zhijie Wu**[*]
Shenzhen University
wzj.micker@gmail.com

**Yang Zhou**[†]
Shenzhen University
zhouyangvcc@gmail.com

**Minglun Gong**
University of Guelph
minglun@uoguelph.ca

**Hui Huang**[†]
Shenzhen University
hhzhiyan@gmail.com

We provide more implementation details of the developed framework [8] in Section 1. Then more experiment results are demonstrated in Section 2 for both qualitative and quantitative comparisons. Finally, we provide the additional stylized examples for high-resolution images in Section 3.

## 1 More discussions on implementation details

In this section, we provide more implementation details about the network architecture.

### 1.1 Revised non-local block

We assemble the non-local block into the bottleneck layer of the proposed framework to capture the long-range dependencies between pixels and make the processed error features more compatible to the current stylized results. The architecture of the revised non-local block is shown in Figure 1. Different from the way used in [9], where all the inputs come from the same image, we feed the top feature of stylized image $f_{in}^4$ and the full error feature $\Delta E^4$ into the block. Then we measure the similarities between the error feature of one pixel and the features for stylized result at other locations to determine what error information to be transited from one pixel to another. In this way, we find that it is effective to capture long-range dependency with the advantages in respecting texture consistency.

Figure 1: The architecture of non-local block.

### 1.2 Training detail

We implement our framework with Tensorflow [1]. In the decoder, nearest-neighbor upscaling plus convolution strategy is used to up-sample features before they are further processed with error propagation blocks. We choose Adam optimizer [3] with a batch size of 4 and a learning rate of 0.0001 for 100000 iterations at each level of a pyramid. Place365 database [11] and WikiArt dataset [5]

---

[*]Equal contribution, in alphabetic order.
[†]Corresponding authors.

Figure 2: Detail close-ups for the ablation study on the effect of a Laplacian pyramid.

Figure 3: More results for effect of a Laplacian pyramid.

are used as the content and style images respectively, following [6]. During training, we resize the smaller dimension of each image to 768 pixels, keeping the original image ratio. Then randomly sampled patches of size $768 \times 768$ are applied for model training. Note that in testing stage, the developed method can process the content and style images with arbitrary size.

## 2 Experiments and Results

### 2.1 Extra ablation study

To demonstrate the effect of a Laplacian pyramid, we implement models with the error computation but dis-enable the pyramid strategy so that we can keep the same resolution during the whole stylization. Both Figure 2 and Figure 3 illustrate that, with error-correction mechanism, all the models are successful in preserving high-level content structure and receiving presented style patterns, like brush strokes, color distributions and block-wise patterns. But with a pyramid, models can better

| Content | w/o block | Ours | Style |

Figure 4: Ablation study on the non-local block. The model without blocks fails to keep style pattern consistency in the sky and land.

Table 1: Ablation study on progressive strategy with a Laplacian pyramid and nonlocal blocks with perceptual metrics. All results are averaged over 100 synthesized images. Note that $K$, $\hat{K}$, $\tilde{K}$ and $K^\star$ all represent the number of refinements. Specifically, $K$ denotes a full model with both Laplacian pyramid and non-local blocks, $\hat{K}$ represents a cascaded model with non-local blocks while the pyramid is not enabled, and $\tilde{K}$ indicates a model that keeps the pyramid but removes non-local blocks at the bottleneck layer. Moreover, $K^\star$ is a model with simple refinements that updates the intermediate stylized results directly with features of input images and keeps the same resolution during the whole stylization process.

| Loss | $K=1$ | $K^\star=2$ | $K^\star=3$ | $\tilde{K}=2$ | $\tilde{K}=3$ | $\hat{K}=2$ | $\hat{K}=3$ | $K=2$ | $K=3$ |
|---|---|---|---|---|---|---|---|---|---|
| $L_c$ | 10.9117 | - | - | 17.4923 | 19.3892 | 13.4950 | 15.8723 | 16.8395 | 18.3771 |
| $L_s$ | 28.3117 | - | - | 16.2673 | 8.9254 | 15.1933 | 9.6819 | 12.1404 | 7.1984 |

capture coarse style patterns and adopt them to different scales thus exhibiting more abundant patterns and superior details. When the number of levels increase, the synthesized image becomes more similar to the style image. According to Table 1, the usage of pyramid strategy improves the style loss as well.

Moreover, the non-local block plays an important role in modeling long-range dependency to respect semantic relationship. We remove this block in the bottleneck layer of our network to verify its importance. As illustrated in Figure 4, our model evidences better texture consistency than the model without non-local blocks.

We also conduct experiments for quantitative comparison regarding perceptual loss and list the results in Table 1. It is shown that, non-local blocks enable us to further improve the fidelity to spatial structures and style patterns. The models with a pyramid achieve better style losses, demonstrating its effectiveness to make scales consistent and introduce fine-grained style details.

## 2.2   More results of our method

In this part, we present some additional stylization results and applications of using the proposed method. Firstly, close-up views for detail comparison and coarse-to-fine generation chain are shown in Figures 5 and 6, respectively.

For a style transfer model, it is important to achieve spatial control by stylize different content image regions with different styles. In Figure 7, masks are applied to define the correspondence between image regions and style patterns. Given one mask, our model is able to achieve the optimal transfer for specified region with mask-out operation.

Table 2: Quantitative results on refinements for existing methods.

| Loss | AdaIN [2] | WCT [4] | Avatar-Net [7] | AAMS [10] |
|---|---|---|---|---|
| Content($L_c$) | 11.4325 | 19.6778 | 15.5150 | 14.2689 |
| Style($L_s$) | 71.5744 | 26.1967 | 42.8833 | 40.2651 |
| Refined Content($L_c$) | 17.2510 | 19.5373 | 19.3490 | 19.2336 |
| Refined Style($L_s$) | 44.7025 | 14.6084 | 13.3498 | 21.2741 |

More refined stylization results for existing methods are present in Figure 9. As we can see, our model is successful in improving the stylization level by making the color distribution closer to the style image and generating more noticeable brush-stroke patterns. We also conduct a quantitative experiment to measure how much improvement the refinement achieves. Based on 100 synthesized images, we load the refined results into the loss module and compute both content and style losses. We present the quantitative results in Table 2, compared to existing methods, our model achieves significant improvements, clearly proving the effect of developed error-correction procedure.

Moreover, we form one matrix of style transfer results in Figure 8. Our model is good at respecting the semantic structures while preserving the target styles like color distribution and texture patterns.

## 3 High-resolution Stylization

Here we prove the ability of our proposed model to transfer styles for high-resolution images. In Figures 10, 11 and 12, we compare our method with WCT and Avatar-Net. As we can see, our results exhibit better texture details adaptive to the spatial structures of content images. In contrast, both WCT and Avatar-Net blur the content spatial structures and introduce unseen style patterns. Moreover, our model also keep the texture consistency within different regions better than those two methods.

## Acknowledgement

We thank the anonymous reviewers for their constructive comments. This work was supported in parts by NSFC (61861130365, 61761146002, 61602461), GD Higher Education Innovation Key Program (2018KZDXM058), GD Science and Technology Program (2015A030312015), Shenzhen Innovation Program (KQJSCX20170727101233642), LHTD (20170003), and Guangdong Laboratory of Artificial Intelligence and Digital Economy (SZ).

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

Figure 5: Detail comparison. The top row shows close-up views of highlighted areas for a better visualization.

Figure 6: Detail close-ups for the coarse-to-fine generation chain.

Figure 7: Application for spatial control. Left: content image. Middle: style images with masks to indicate target regions. Right: synthesized result.

Figure 8: Stylization matrix of applying different styles to different content images.

Figure 9: Refinements for existing state-of-the-art methods. Our model improves the stylization level in terms of color distribution and brush-stroke patterns. Note that the 3rd column indicates the close-up views for results of existing methods while the 4th column represents the stylized images refined by our method.

AvatarNet

WCT

Ours

Figure 10: Comparison on high-resolution stylization. Note that the resolution of the content image in this figure is $1024 \times 1024$.

AvatarNet

WCT

Ours

Figure 11: Comparison on high-resolution stylization. Note that the resolution of the content image in this figure is $2048 \times 1280$.

AvatarNet

WCT

Ours

Figure 12: Comparison on high-resolution stylization. Note that the resolution of content image in this figure is $2048 \times 1280$.