[Reviews · NeurIPS 2019]

Reviewer 1



-- I am impressed with the improvement, in terms of visual quality, compared to previous approaches. The visual examples have lots of details and preserves high-level content structure better. -- The idea is intuitive. I am not aware of other similar approaches, to the best of my knowledge.

Reviewer 2



(1) Originality: This paper first introduces error-correction and diffusion mechanism into the style transfer literature, which separates it from existing works. Meantime, style transfer via iterative refinement is not a novel idea as it has been applied by the WCT method [17]. (2) Quality: This paper has provided both qualitative and quantitative experiments to show the superiorities of the proposed style transfer method. However, there are several concerns: a. The main concern is the meaning of equation (3). Equation (2) associates the style and content error via a learnable weight W to obtain a full error feature. In equation (3), the affinity is calculated between this full error feature and the stylized image features. As far as I can see, these two features have different meanings and are not comparable to each other. So what is the motivation to calculate the affinity between them? How would a further multiplication between this affinity and the full error feature diffuse the error to the whole image? b. In equation (4), why not simply concatenate the error feature of layer i with error feature of layer i-1, why using a fusion layer with learnable weight? c. As for the performance of the proposed algorithm. First, I do not think 0.5680 second per image (Table 2) is real-time (as stated in the introduction). Second, this work has to separately train each error transition network separately, that would increase training burdens compared to other methods such as AdaIn or WCT. However, the proposed algorithm does achieve lower style loss and the results look better compared to other state-of-the-art methods. (3) Clarity: In general, this paper is easy to follow but contains some spelling mistakes, e.g., Line 123, inputted -> input Line 109, 124, outputted -> output There are also some confusions in equations. Such as the ‘*’ symbol in equation (3), I suppose it to be an element-wise multiplication, which is different from the matrix multiplication in equation (2), which the author(s) should make clear. (4) Significance: This paper aims at improving the existing style transfer method via interactive error-correction. The framework is novel compared to other style transfer methods. Post rebuttal: The rebuttal indeed does not clearly answer my question. However, after revisiting the details of the paper, and inferring from the comments of other reviewers, I could get the insights, i.e., not strictly, a forward way of optimizing the transferred image, with the attention mechanism involved. Thus I raised the rating.

Reviewer 3



The idea of self error correction is smart. It requires the computation of style/content features in multiple resolution. It would be nice to discuss the relation with iterative back-propagation.

[Author Response · NeurIPS 2019]

Thanks for the careful and valuable comments from reviewers. We are glad to see the agreements that our paper proposes a new approach (R1), and the idea of error-correction mechanism is intuitive (R1), novel (R2) and smart (R3). Compared to previous methods, impressive visual improvement is achieved (R1) and our results look better both quantitatively and qualitatively (R2). Reviewers have some concerns about the motivation of affinity matrices in non-local blocks (R2), the relation and comparison with iterative back-propagation method (R3) and some other details. However, we believe all these comments can be addressed through this rebuttal and a minor revision, which will include some of the explanations below as well as spelling corrections.

**Q1.** *Is any special feature operation applied in ETN? & Does a larger K help? (R1)*

A1. Please note that, except employing the learnable matrices to fuse error features, our model is built based on some conventional operations, like convolutions. And as presented in Line 148 of submission, subtle improvement is achieved with larger K for images smaller than $2K \times 1K$.

**Q2.** *The motivation to compute affinity matrices & How to achieve the error diffusion. (R2)*

A2. We respectfully point out that both stylized image features and corresponding full error features encode information from the same content-style image pairs and can definitely correlate with each other. If we simply apply full error features to update stylized images, similar to Gatys et al. [7], it is easy to encounter local minima and hurt semantic structures. Thus as mentioned in Line 35 of the submission, our goal is to extract more compatible error features for refinements. We assume that when the error feature is more correlative to the feature of stylized result, they are more compatible to each other. Then for a stylized image, an affinity matrix can be used to measure its similarities to error features at different pixels. Thus similar to [23, 11], to better preserve the long-range dependency between pixels, we diffuse errors to the whole image through a matrix multiplication between a full error feature and an affinity matrix. Please see Fig. 5 in submission for example. Without joint analysis strategy, the model stylizes images with unseen white bumps and blurs the structure of clouds more. And as shown in Fig. 3 in supp file, our full model equiped with non-local blocks can produce results with clearer outlines (e.g., the edges between the sky and mountain) and better respect the semantic information while the model without blocks fails to preserve the orientation of the road in land. Moreover, as shown in the Table 1 in supp file, the non-local blocks can also improve the perceptual content & style metrics, making the diffusion mechanism effective.

**Q3.** *Why not simply concatenate the error feature of layer i with error feature of layer i-1? (R2)*

A3. Feeding $\Delta E^i$ into convolution layers, we denote $\Delta \hat{E}^i \in R^{N \times C^{i-1}}$ as the processed feature, $\Delta \hat{E}^i = \Delta E^i \otimes \phi_t$. N and $C^{i-1}$ indicate the number of pixels and the channel number of $\Delta \hat{E}^i$ respectively. Thus we clarify that we apply the learnable matrix to merge $\Delta \hat{E}^i$ with $\Delta E_s^{i-1} \in R^{C^{i-1} \times C^{i-1}}$ due to their different dimensions.

**Q4.** *Performance issues, including increased training burden and running time. (R2)*

A4. Thanks for pointing out the mistake in real-time stylization, which will be corrected in revision. Our method indeed slightly increases training burdens, but achieves remarkable improvements with comparable running time.

**Q5.** *Relation and comparison with iterative back-propagation method. (R3)*

A5. We clarify that we share the same stylization goal and content & style definition as Gatys et al. [7]. However, one key difference is that [7] back-propagates the error features while our method achieves the error-correction in a feed-forward manner. Thus we can explicitly perform joint analysis between errors and a synthesized image for a better residual image and apply the error diffusion to capture the long-range dependency in pixels. Moreover, the developed technology can be extended to other fields more easily thanks to its efficiency and effectiveness. Fig. 1(a) shows that, [7] often gets stuck into local minimum and fails to capture the salient style patterns. In comparison, our approach, shown in Figs. 6-7 of the submission, yields more visually pleasing results.

**Q6.** *Comparison with ResNet like structure as an alternative to the structure in Figure 2. (R3)*

A6. Laplacian pyramid can facilitate performance in various fields [1, 9, 5]. In fact, we have built a baseline model by keeping the same resolution during stylization and demonstrate the results in Fig. 2 and Table 1 of supp file and Fig. 1(b). With error-correction mechanism, the baseline model is successful in preserving high-level content structure and receiving presented style patterns, like brush strokes, color distributions and block-wise patterns. But the pyramid strategy enables us to better capture large-scale style information and communicate different scales, thus exhibiting more abundant patterns and superior details. The usage of pyramid also improves the style loss.

Figure 1: Stylization results of Gatys et al. (a) and ResNet-like baseline (b) for tests shown in Figs. 6-7 in submission.

[Meta-Review · NeurIPS 2019]

All reviewers were positive towards the paper. They found the iterative solution and the introduction of self error correction to be an interesting addition to the Style transfer toolkit. The method was also considered to produce good results. While two of the reviewers only recommended acceptance marginally, there is consensus that the paper should be published.